# The Energy Contents of Broken Rice for Lactating Dairy Cows

**DOI:** 10.3390/ani13193042

**Published:** 2023-09-27

**Authors:** Thidarat Gunha, Kanokwan Kongphitee, Bhoowadol Binsulong, Kritapon Sommart

**Affiliations:** 1Department of Animal Science, Faculty of Agriculture, Khon Kaen University, Khon Kaen 40002, Thailand; thidarat.gunha@kkumail.com (T.G.); b.bhoowadol@kkumail.com (B.B.); 2Department of Applied Biology, Faculty of Sciences and Liberal Arts, Rajamangala University of Technology Isan, Nakhon Ratchasima 30000, Thailand; kanokwan.aof1@gmail.com

**Keywords:** ruminant, broken rice, digestibility, calorimetry, energy requirement

## Abstract

**Simple Summary:**

The net energy for lactation of feedstuffs is critical data for a precision dairy feeding system aimed at improving animal productivity and sustainability. The energy value of feedstuffs is required to determine the energy requirements. Broken rice is a byproduct of rice (*Oryza sativa*) processing that can be used as an energy feed for livestock. In research on broken rice for lactating dairy cows in the tropics, data on energy utilization still needs to be included. Evaluation of the net energy content of broken rice will allow for more precise feeding for lactating dairy cows. Therefore, we aimed to determine the net energy content of broken rice for lactation using animal calorimetry. The findings indicate that increasing the amount of broken rice in cows’ diets could improve dry matter, organic matter, and fiber digestibility but not adversely affect intake, energy balance, or production performance. The net energy for lactation of broken rice in dairy cows was estimated at 8.68 MJ/kg.

**Abstract:**

This study aimed to evaluate (1) the net energy for lactation of broken rice in dairy cows and (2) the effects of broken rice substituting in diets on feed intake, nutrient energy utilization, and milk production. An energy metabolism experiment was conducted using a respiration chamber system in four multiparous Holstein crossbred cows (88.6% Holstein × 11.4% Native Thai; body weight of 438 ± 16.0 kg; 70 ± 31 days in milk) according to a 4 × 4 Latin square design with four 21-d periods. The four dietary treatments included a basal diet substitution with broken rice at 0%, 12%, 24%, and 36%. Increasing the substitution rate of broken rice in the diet resulted in unaffected feed intake, milk yield and composition, and energy balance (*p* > 0.05); however, a linear increase in the digestibility of dry matter, organic matter, and neutral detergent fiber (*p* < 0.05). The estimated net energy for lactation of broken rice was 8.68 MJ/kg. The net energy requirement for maintenance was estimated at 504 kJ/kg of metabolic body weight. Our results indicated that broken rice is a good energy-feed resource and that increasing the proportion in the diet up to 36% had no adverse effect on dairy cows’ production performance.

## 1. Introduction

The evaluation of innovative feed ingredients is essential for the long-term improvement of livestock productivity. The most common method to determine the energy values of individual feedstuffs is open-circuit whole animal respiration chambers, in which heat production is indirectly measured using respiratory gas exchange and methane emission [1]. Determining the energy value of feedstuffs will allow for a more accurate diet formulation for lactating dairy cows [1,2,3]. Recently, Foth et al. [2] determined that the energy content of reduced-fat dried distiller grains with soluble metabolizable energy (ME) was 14.3 MJ/kg, and the net energy for lactation was 8.5 MJ/kg. Gunha et al. [3] estimated that the net energy for lactation of cassava chips was 8.0 MJ/kg for crossbred dairy cows. In addition, Wei et al. [4] estimated that the net energy value of rice straw was 3.4 MJ/kg for beef cattle using the regression method by indirect calorimetry.

Rice (*Oryza sativa* L.; long-grained variety) is the seed of an annual plant in the Gramineae family that is one of the world’s most widely planted grain crops, with 510.8 million tons produced worldwide [5]. Rice byproducts are abundant agricultural products, including straw, bran, and broken rice [6]. Nutritional values of broken rice have dry matter (DM) of 86.9% to 93.6%, ash, crude protein (CP), ether extract (EE), neutral detergent fiber (NDF), acid detergent fiber (ADF), and starch within the ranges of 0.4% to 5.0%, 3.1% to 10.2%, 0.5% to 2.2%, 0.8% to 9.3%, 0.6% to 2.5%, and 70.8% to 87.8% of DM, respectively [7,8,9,10,11,12,13]. The gross energy ranges from 14.7 to 15.6 MJ/kg [11,12,13] and ME 7.9 to 12.7 MJ/kg [7,9,10,12]. Therefore, broken rice has excellent nutrient quality and is a locally available year-round feed resource in Thailand [7,9,10]. Kotupan and Sommart [7] showed improved intake and digestibility, total volatile fatty acids, propionate concentration, marbling score, and carcass characteristics when replacing 32% of cassava chips with broken rice in the final phase of fattening beef cattle. Recent research showed that substituting 20% corn with rice grains in the diet of Hanwoo steers did not adversely affect rumen fermentation and growth performance [14]. Miyaji et al. [15,16] reported that substituting corn with rice (*Oryza japonica*, short-grain rice type) has little effect on feed intake, ruminal pH, and milk production while improving dietary nitrogen utilization when cows were fed diets that contained 30.9% to 31.2% rice grain. Miyaji et al. [17] found that substituting steam-flaked corn with rice (30.9%) did not alter milk yield but increased milk fat production. Scheibler et al. [18] found that replacing corn with rice at 0%, 33%, 63.67%, and 100% in dairy cow diets had no adverse effect on animal health, feed intake, digestibility, or milk production. To our knowledge, the use of long-grain varieties of broken rice in the diets of lactating dairy cows is limited. Moreover, information on the net energy value for lactation is lacking. The determination of the net energy for lactation of broken rice will allow for more precise dairy cow feeding systems.

Therefore, the objectives of this study were to evaluate (1) the net energy for lactation of broken rice and (2) the effects of substituting broken rice in diets on dairy cow performance.

## 2. Materials and Methods

### 2.1. Cows, Experimental Design and Diets

The Animal Ethics Committee examined and approved the experimental protocols for the ethical use of animals in research (No. IACC-KKU-10/63).

In this study, four multiparous crossbred cows (88.6% Holstein), with initial body weights (BW) of 438 ± 16 kg (mean SD), were 70 ± 31 days in milk and had a milk yield of 16.9 ± 3.5 kg/d at the start of the experiment. Concrete feed bunks and automatic water troughs were always available for cows. In addition to receiving ad libitum feedings of total mixed ration silage twice daily at 08:00 and 16:00 h, cows were milked in pens twice daily at 06:00 and 15:30 h.

Cows were randomized to receive 1 of 4 dietary treatments, alternating across four periods, according to a 4 × 4 Latin square experimental design. Each experimental period consisted of 21 days of data collection and concluded with five days (d 17 to d 21) of daily measures of respiratory gas exchange, milk, feces, and urine. Cows were kept separately in 2.5 m × 4.5 m pens, holding pens next to one another during the experiment, with free feed and drinking water access.

Three test diets and one basal diet (0% DM of broken rice) were among the four dietary treatments (Table 1). Broken rice was substituted for the three test diets at 12%, 24%, and 36% DM in the basal diet (Table 1). Substitution and regression methods to determine diets’ digestible, metabolizable, and net energy content were applied according to Gunha et al. [3] methods. A regression approach was based on multiple-point substitution, and the substitution and regression methods are necessary to determine individual feedstuffs’ energy content [4]. As a result, various broken substitution ratios into basal diets are used in this study.

A 5000 kg capacity horizontal mixer (Celikel TMR feed mixer, 108 Agriculture Machine and Equipment Co., Ltd., Lop Buri, Thailand) was used to mix the ingredients for a total mixed ration (Table 1). Dietary treatment mixture was mixed to a weight of about 2000 kg, then a 35 kg silage bag was packed into high-density polyethylene bags (25 in width × 44 in length, Kwankhawpanich, Nakhon Ratchasima, Thailand), vacuum compressed (Hitachi 1600 W model CV-930F, Hitachi Consumer Products Co., Ltd., Prachinburi, Thailand), and covered with a polypropylene woven bag (23 in width × 37 in length, Thailand). These silage bags were left at ambient temperature (26 to 39 °C) until feeding time.

### 2.2. Feed Intake and Digestibility

The cows were weighed on each experimental period’s first and last days to calculate their DM intake and energy partition as a metabolic weight (BW^0.75^). The daily dietary intake of each animal was determined by subtracting the quantity of feed refused from the quantity offered throughout the collecting period.

Using the Suzuki et al. [19] method, the total collection technique was carried out for each animal for 5 days in a digestion trial pen. We measured and sampled the amount of feces (1 kg) and urine (120 mL) daily. Excreted feces were immediately collected in pans positioned behind the cow, weighed, and sampled for digestion analysis. The total urine was contained in a tube urine cup and kept at a pH below 3.0 in plastic tanks with 6 mL of normal hydrochloric acid. Daily feces and urine samples were stored at 4 °C. Following the end of the period for collecting metabolic data, aliquot samples (1 kg) of feed, feces, and urine were composited and stored at −18 °C until analysis.

### 2.3. Animal Calorimetry

The respiratory gas measurement was conducted to determine the oxygen (O_2_) consumption, carbon dioxide (CO_2_), and methane (CH_4_) production of each dairy cow, as described by Suzuki et al. [19]. The respiration chamber system consisted of four units: a digestion trial pen, head cage, gas sampling and analyzing unit, and data recording and processing unit. A head cage is installed in front of the digestion trial pen and designed to be airtight, except for an air-inlet adjustable collar. The cow’s head stayed in the cage zone all day and had access to feed and automated water. Cows can lie down on rubber mats. The airflow flow rate and volume in the respiration chamber were measured using an airflow meter (Nippon Flow Cell, Tokyo, Japan). The O_2_, CO_2,_ and CH_4_ concentrations in inflow and outflow tubes were analyzed using a multi-gas analyzer (MultiExact 4100 Analyzer, Servomex Group Ltd., East Sussex, UK). The standards containing gases included two O_2_ concentrations (18.90% and 24.96%), 1.79% of CO_2_, and 1760 ppm of CH_4_, which were calibrated daily using certified gases (Linde (Thailand) Public Co. Ltd., Samutprakarn, Thailand). The gas recovery test was conducted using the CO_2_ injection method, with values ranging from 98% to 104%. During the last three days of each period, gas data measurements were taken at 7.5 min intervals for 23.30 h (from 08:00 a.m. to 07:30 a.m.) and finally adjusted to 24 h to measure total respiratory gas exchange. Ambient temperature and relative humidity were not conditioned; their average was 27.7 °C (23.3 to 36.6 °C) and 82.7% (41.0 to 99.0), respectively. The digestible energy (DE) intake was deducted from the urine and methane energy to determine the ME intake. Total heat production (HP) was calculated using the Brouwer method [20].

### 2.4. Sampling and Chemical Analysis

Feeds were sampled to determine the chemical composition and fermentation quality. The dry matter (DM) content was determined using a fan-forced oven set at 105 °C, reaching a constant weight. Feeds and feces nutrient content were analyzed using the protocols described in AOAC methods [21] for DM (method 967.03), ash (method 942.05), CP (method 984.13), and EE (method 920.39). The fiber (NDF, ADF) concentrations were determined with a fiber analyzer (ANKOM 200/220, ANKOM Technology, Macedon, NY, USA), and then NDF was treated with sodium sulfite and alpha-amylase [22]. The equation used to calculate the non-fiber carbohydrate (NFC) content was NFC (%) = [100 − (%CP + %NDF + %EE + %Ash)]. Gross energy content was determined using a bomb calorimeter (IKA C2000 Basic, IKA-Werke, Staufen, Germany).

To determine the fermentation profile of the silage, a fresh silage sample (20 g) was blended with distilled water (180 mL) and stored in a refrigerator at 4 °C for 24 h. After that, the extracts were passed through a nylon filter funnel. The silage juice determined pH, ammonia nitrogen, lactic acid, and short-chain fatty acid concentrations. The pH was measured immediately with a bench pH meter (Eutech pH 700, Eutech Instruments Pte Ltd., Ayer Rajah Crescent, Singapore). Lactic acid and volatile fatty acid concentrations were analyzed using gas chromatography (GC2014, Shimadzu, Tokyo, Japan) [23].

Milk production was recorded daily, and milk samples were collected at milking times (06:30 h and 15:30 h) for five consecutive days from days 17 to 21 of each period. Two bottles (110 mL/bottles) were collected for each milking, and one aliquot was stored refrigerated at 4 °C until milk composition analysis was analyzed using the milk composition analyzer (MilkoScan^TM^ 7RM, Foss Electic, Hillerod, Denmark) and somatic cell count using Fossomatic^TM^ 7 DC (Fossomatic DC, Hillerod, Denmark).

### 2.5. Calculation

Fat and protein-corrected milk (FPCM, kg/day) was calculated as illustrated in Equation (1) according to Gerber et al. [24]:FPCM = (0.337 + 0.116 × fat% + 0.06 × protein%) × milk yield (kg/day)(1)

Energy-corrected milk (ECM, kg/day) was calculated as illustrated in Equation (2) according to Cabezas-Garcia et al. [25]:ECM = (milk yield (kg/day) × milk energy (MJ/kg))/3.1(2)

Milk energy (MJ/kg) was calculated as illustrated in Equation (3), according to Cabezas-Garcia et al. [25]. Milk fat, protein, and lactose units are expressed as g/kg.
Milk energy = (0.0384 × fat) + (0.0223 × protein) + (0.0199 × lactose) − 0.108(3)

Heat production (HP, kJ/day) was calculated by using the volumes of respiratory gas (O_2,_ L/day, CO_2_ L/day, CH_4_ L/day) and urinary nitrogen (UN) excretion (g/day) according to Brower [20], as shown in Equation (4):HP = 16.18 × O_2_ + 5.02 × CO_2_ − 5.99 × UN − 2.17 × CH_4_,(4)

Methane energy (kJ/day) was estimated by multiplying CH4 production (L/day) by a constant of 39.54 [26]. Energy balance (EB; kJ/kg BW^0.75^) was determined using Equation (5).
EB = ME intake − HP − Milk energy(5)

The efficiency of metabolizable energy use for lactation (*k*_l_) was calculated according to a previous report [27,28] using Equation (6).
*k*_l_ = E_l(0)_/(ME intake − ME_m_)(6)
where E_l(0)_ is the milk energy output (E_l_) adjusted to zero energy balance (MJ/day) and calculated from Equations (7) and (8). ME_m_ is the ME requirement for maintenance (MJ/day).
EB > 0, E_l(0)_ = E_l_ + 0.95 × EB(7)
EB < 0, E_l(0)_ = E_l_ − 0.84 × EB(8)

DE, ME, and NE_L_ contents of test ingredients were calculated as illustrated in Equation (9) according to Gunha et al. [3]:E_ti_ (MJ/kg DM) = [E_td_ − (1 − P_ti_) × E_bd_]/P_ti_(9)
where E_ti_ (MJ/kg DM) = the energy content of the broken rice test ingredient, E_td_ (MJ/kg DM) = the energy content of the test diet, E_bd_ (MJ/kg DM) = the energy content of the basal diet, and P_ti_ = the ratio of test ingredient substitution in the basal diet.

### 2.6. Statistical Analysis

Regression equations were generated using the REG procedure of SAS version 9.0 [29]. Linear regressions of broken rice-associated energy intake (dependent variables) against testing ingredient intake (independent variables) were conducted to determine digestible, metabolizable, and net energy for lactation of broken rice [4].

All experimental data were subject to analysis of variance using the GLM procedure of SAS [29] for a 4 × 4 Latin square design model:Y_ijk_ = μ + ρ_i_ + γ_j_ + τ_k_+ ε_ijk_,
where Y_ijk_ was the response of the dependent variable, μ was the observation means, ρ_i_ was the random effect of the period (i = 1 to 4), γ_j_ was the random effect of the cow (j = 1 to 4), τ_k_ was the fixed effect of treatment (k = 1 to 4), and ε_ijk_ was the residual error. Orthogonal polynomial analysis was conducted to determine the linear, quadratic, and cubic effects of dietary treatment. Significance was declared at *p* ≤ 0.05.

## 3. Results

Fifteen of the possible 16 energy balances were completed. During the data collection of the third period, one cow was removed from total tract digestibility, and respiratory gas collections became ill from hardware disease. The consulted veterinarian treated the hardware disease, including antibiotics, and administered a magnet into the rumen. The cow was used for data collection in the fourth period after feed intake and milk yield recovered well by 12 days.

### 3.1. Diet Composition

The chemical composition, fermentation quality of broken rice, and diets are presented in Table 2. All the dietary treatments were low in pH, VFA, and NH_3_-N but had high lactic acid content, indicating that all the diets were good-quality silages.

### 3.2. Feed Intake and Digestibility

The intake of DM and OM was unaffected (*p* > 0.05) by dietary treatment. The substitution of broken rice in the dietary treatment resulted (Table 3) in a linear decrease in CP intake, EE intake, ADF intake, and NDF intake (*p* < 0.05). In contrast, the intake of NFC increased linearly (*p* = 0.02).

Apparent digestibility is shown in Table 4. The digestibility of CP, EE, ADF, and NFC was unaffected (*p* > 0.05) by diets. Increasing the substitution of broken rice in the diet resulted in a linear increase (*p* < 0.05) in the apparent digestibility of DM, OM, and NDF.

### 3.3. Milk Production

Increasing the substitution of broken rice in the diets did not affect the milk production or the composition of milk protein, fat, lactose, SNF, energy, or SCC (*p* > 0.05). The feed efficiency was unaffected by the dietary treatments (*p* > 0.05) (Table 5).

### 3.4. Respiratory Gas Exchange and Energy Partitioning

The respiratory gas exchange analysis is presented in Table 6. Dietary treatment did not affect oxygen consumption, carbon dioxide production, methane emission, or respiratory quotient (*p* > 0.05).

Energy partitioning, energy intake, energy content, and energy utilization are shown in Table 6. The energy lost as feces, urine, methane emission, heat production, and energy balance were unaffected by treatment. In addition, milk energy, E_l(0)_, energy intake, and energy content were unaffected by the broken rice in the diet (*p* > 0.05). Energy utilization efficiency was unchanged (*p* > 0.05) by the broken rice substitution in the diet.

### 3.5. Maintenance Energy Requirement and Efficiency of ME Utilization for Lactation

Estimations of the maintenance energy requirements of lactating crossbred dairy cows by linear regression are shown in Figure 1. Regression (E_l(0)_ = 0.756 × ME intake − 504.3) was highly significant (*p* < 0.001), and the R^2^ value was 0.75. The ME_m_ of Holstein crossbred dairy cows was 667 kJ/kg BW^0.75^, and the efficiency of ME utilization for lactation was 75.6%. As derived, the net energy requirement for maintenance (NE_m_) was 504.3 kJ/kg BW^0.75^.

### 3.6. Energy Content for Lactation of Broken Rice

The regressions of broken rice-associated energy intake against the broken rice substitution amount are shown in Table 7. The equation for the DE was *Y* = 13.13*X* + 0.02 (R^2^ = 0.98, RMSE = 0.25, *p* < 0.0001, *n* = 15), and the ME was Y = 11.87*X* + 1.91 (R^2^ = 0.96, RMSE = 0.31, *p* < 0.0001, *n* = 15). Broken rice has an energy value of 13.13 MJ DE/kg and 11.87 MJ ME/kg. In addition, the NE_L_ regression equation was *Y* = 8.68*X* + 1.18 (R^2^ = 0.85, RMSE = 0.50, *p* < 0.0001, *n* = 15), and the net energy value for broken rice was determined to be 8.7 MJ NE_L_/kg.

The DE, ME, and NE_L_ values (Table 8) of broken rice derived were similar between the single substitution and regression methods (*p* > 0.05). The energy values of broken rice, estimated by the substitution method, were 13.06 MJ DE/kg, 11.81 MJ ME/kg, and 8.68 MJ NE_L_/kg, respectively.

## 4. Discussion

### 4.1. Nutrient Intake and Milk Production

Feed intake and digestibility are essential limiting factors determining nutrients and energy supply for maintenance and production. Tropical feed resources are often low-quality and deficient in digestible nutrients and energy intake. There are generally few milk-producing cow herds in Thailand and other tropically developing countries. In Thailand, the average milk yield is approximately 12 kg/day. The limitations are heat stress and quality feed supplies. Increasing the available nutrient supplies enables animals to improve milk production from carbohydrates, proteins, and fats. In the present study, the proportion of broken rice substitution in the basal diet increased, and the content of non-fiber carbohydrates increased, whereas the contents of CP, NDF, ADF, and EE decreased. The silage fermentation qualities were characterized by low pH (<3.9) and high lactic acid content after seven days of ensiling, indicating that all the diets were good-quality silages, like a previous report by Gunha et al. [3], Kotupan and Sommart [7], Wang and Nishino [30], and Kongphitee et al. [31]. The level of broken rice substitution in the basal diet of dairy cows did not affect DM and OM intake, which is consistent with the findings of Kotupan and Sommart [7], who reported on fattened beef cattle. Our data indicates that non-fiber carbohydrate intake substantially improved, suggesting a significant increase in total digestible nutrients and energy supply to the cows when substituting broken rice. Similarly, Miyaji et al. [15] fed lactating cows a diet by replacing corn with rice at 0%, 15.6%, and 31.2% and observed no differences in dry matter intake or milk production.

This study analyzed chemical analysis (Table 2). Total dietary NDF ranges from 33.7% to 45.8%, with 21.5% to 22.8% total forage NDF concentration, indicating that the diets provided sufficient dietary NDF fiber requirements for dairy cows. The NRC [32] recommended minimum concentrations of total dietary NDF for cows at 25%, with the condition that 19% of dietary NDF is from forage. In this study, rice straw was mainly the total forage NDF concentration contribution calculated, which ranged from 21.5% to 22.8%; therefore, it may be expected that the physical structure of forage could maintain chewing, ruminal, and cow health. Our previous work has suggested that 10% of rice straw included in the total mixed ration of native Thai cattle and Holstein crossbred bulls maintained an average daily 8.5 to 9.85 h chewing time and 3.1 to 4.7 h ruminating time, suggesting that most tropical feed and systems depend on agricultural industry by-products such as rice straw, cassava pulp, and palm kernel cake that have a high NDF and indigestible NDF fraction that may be contributing to the rumen floating mat and stimulating sufficient ruminating activities [31,33].

In this study, there was no difference in dry matter intake or milk yield among the dietary treatments. The assignment of four cows per treatment according to a 4 × 4 Latin square design in this study may be inadequate to detect treatment differences, resulting in no differences in dry matter intake or milk yield. Most of the variability in the research can be attributed to high between-cow variability [34]. In the future, the sample size should be increased to reduce the variability and increase the reliability of the in vivo feeding experiment. Our data indicate that neither milk protein, lactose, and SNF yield nor milk composition of protein, fat, lactose, and SNF were affected by dietary treatments with an increasing broken rice level of up to 36% of DM in the diet. Similarly, Miyaji et al. [15,16] reported that replacing corn with brown rice did not affect milk yield. Miyaji et al. [15,16,35] also found that substituting brown rice for corn did not affect milk fat concentrations. In contrast, Miyaji et al. [35] reported that the milk yield decreased when substituting brown rice for corn in diets with a high proportion of grain (>40% of DM) had adverse effects on lactating cow productivity.

### 4.2. Energy Partition, Efficiency of Metabolizable Energy Utilization

There was no difference among dietary treatments in respiratory gas, energy partitioning, energy intake, energy content, energy utilization, or enteric methane emissions when broken rice was substituted in the diets. Enteric methane is a natural methanogen fermentation product of carbohydrates and amino acids in the rumen and hindgut. The organic substrates in the diets are ruminal fermented, producing short-chain fatty acids and H_2_, from which H_2_ and CO_2_ synthesize methane by rumen methanogen. Our result indicated that alternating the diet composition with broken rice did not affect enteric methane emission energy loss (kJ/kgBW^0.75^). In this study, daily methane emissions (258.7 to 303.6 L/d) were within the range previously reported by Gunha et al. [3], Suzuki et al. [19], Kongphitee et al. [31], and Binsulong et al. [33] of cattle in the tropics, ranging from 146 to 360 L/d.

Estimating the metabolizable energy requirement for maintenance (ME_m_) value of 667 kJ/kg BW^0.75^ for Holstein crossbred dairy cows was achieved through regression of the daily milk energy yield against the metabolizable energy intake (Figure 1). These values of ME_m_ were higher than in Bos taurus × Bos indicus dairy crossbreds (558 to 599 kJ/kg BW^0.75^) reported by Olivera et al. [36] and Binsulong et al. [33]. However, the ME_m_ in this study was lower than the values for Holstein (710 kJ/kg BW^0.75^) and Holstein crossbred dairy cows (670 kJ/kg BW^0.75^) [37,38]. The efficiency of metabolizable energy utilization for lactation value in this study was 0.76, which agreed with a similar value of 0.76 to 0.88 reported by Foth et al. [2], Judy et al. [39], and Binsulong et al. [33]. Recent estimates of *k*_l_ have varied from 0.50 to 0.81 [32]. The varied energy efficiency and requirements were primarily attributed to the animal breed, physiological state, feed intake, and environmental conditions [32,36,37,38].

### 4.3. Net Energy for Lactation Estimation Methods

In estimating the net energy values for feedstuffs, the substitution method was widely accepted by feeding the single point of test feed in the basal diet [4]. Compared with the regression method based on multiple-point substitution, the energy values derived are more robust than single-point substitution [4]. The multiple-point substitution regression method has recently been used to estimate the energy values of ruminants. The net energy for lactation of broken rice in animal calorimetry has not been reported. In this study, the estimated ME and NE_L_ values for the broken rice determined using the substitution method averaged 11.8 and 8.6 MJ/kg of DM, respectively, such as the ME and NE_L_ values of 11.9 and 8.7 MJ/kg of DM when determined using the regression method, respectively. These estimates of ME were 6.3% (12.7 MJ/kg of DM) lower than those reported for broken rice in vitro by Kotupan and Sommart [7] but 15 to 33% higher than those reported for ruminants using an in vitro technique by Chumpawadee et al. [9] and Nitipot and Sommart [10] (7.9 and 10.1 MJ/kg of DM, respectively). Our result was within the range of the recent estimates of net energy for lactation for reduced-fat dried distiller grains with soluble at 8.5 MJ/kg [2], and Gunha et al. [3] estimated that the net energy for lactation for cassava chips was 8.0 MJ/kg. Our data suggested that the in vitro method had a more significant variation in the estimation of net energy for lactation in dairy cows when compared with the in vivo method. The limitation of animal calorimetry to determine net energy for lactation remained because in vivo experiments require whole animal metabolism studies and respiratory gas measurements; they are time-consuming, costly, and require many feed samples.

The advantage of a 4 × 4 Latin square design is that it allows us to determine the effect of cow, period, and change responses to dietary treatment. However, the short-term measurement (21 d) and number of replication cows are limitations of this study. Dairy cow production performance may be required to be confirmed in the long-term feeding experiment. The limitation of this experiment was also due to the substitution method [4] involving substituting a test feed (broken rice) into each ingredient in the basal diets; consequently, the results on animal performances are not only because of the broken rice but also because of ingredient composition change.

## 5. Conclusions

Our study, utilizing tropical Holstein crossbred cows, evaluated the net energy for lactation of broken rice and the effects of the substitution of broken rice on dairy cows’ performance. The results indicate that substituting broken rice up to 36% in the dairy diet did not affect dry matter intake or milk production but showed a linear increase in the digestibility of dry matter, organic matter, and neutral detergent fiber. Using the in vivo animal calorimetry method, the net energy for lactation of broken rice predicted was 8.68 MJ/kg. The metabolizable energy requirement for maintenance in crossbred dairy cows was 504 kJ/kg BW^0.75^, and the efficiency of metabolizable energy used for lactation was 0.76. The energy values of broken rice derived from substitution and regression methods were similar. Long-term feeding experiments are needed to develop dairy cattle-fed broken rice feeding systems.

## Figures and Tables

**Figure 1 animals-13-03042-f001:**
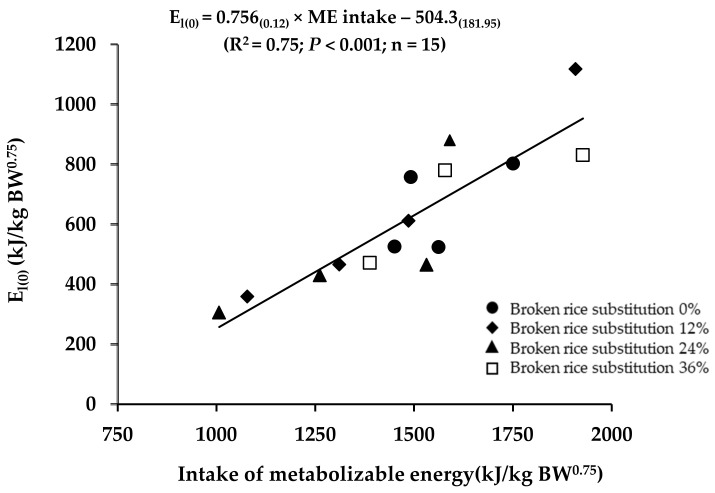
Regressions of adjusted milk energy output to zero energy balance (E_l(0)_, kJ/kg BW^0.75^) and the intake of metabolizable energy (kJ/kg BW^0.75^) in lactating dairy cows fed the diet with different substitutions of broken rice.

**Table 1 animals-13-03042-t001:** Ingredients, feed formulation, and costs of the four broken rice dietary treatments.

	Broken Rice Substitution Ratio (%)			
Items ^1^	Basal Diet (0)	12	24	36
Ingredients, %				
Rice straw	15.0	13.2	11.4	9.6
Broken rice *	0.0	11.8	23.5	35.3
Cassava pulp	30.0	26.4	22.8	19.2
Wet brewery waste	15.0	13.2	11.4	9.6
Rice bran	11.0	9.7	8.4	7.0
Palm kernel cake	12.0	10.5	9.1	7.7
Soybean meal	15.0	13.2	11.4	9.6
Urea	0.5	0.5	0.5	0.5
Mineral ^1^	1.0	1.0	1.0	1.0
Premix ^2^	0.5	0.5	0.5	0.5
Total	100.0	100.0	100.0	100.0
Cost (Thai Bath/kg FM)	3.64	4.43	4.79	5.47

* Cost 12.33 TBH/kg FM. ^1^ Minerals (Mineral #0106410029, Dairy Farming Promotion Organization of Thailand, Saraburi, Thailand). ^2^ Premix (Golden Mix S # 0104610040, DFC Advanced Co. Ltd., Khon Kaen, Thailand).

**Table 2 animals-13-03042-t002:** The chemical composition and fermentation quality of broken rice and the four broken rice substitution dietary treatments.

		Broken Rice Substitution (%)			
Items ^1^	Broken Rice	0	12	24	36
Chemical composition					
Dry matter, %	88.9	32.7	34.0	36.1	40.8
Organic matter, %DM	99.2	93.6	94.5	95.1	95.4
Crude protein, %DM	7.4	18.5	16.7	15.5	14.5
Neutral detergent fiber, %DM	6.4	45.8	42.2	41.3	33.7
Acid detergent fiber, %DM	1.2	23.5	23.2	20.5	17.1
Ether extract, %DM	1.3	5.9	5.3	5.1	4.5
Non-fiber carbohydrate, %DM	84.9	23.5	30.3	33.1	42.7
Fermentation profile					
pH		3.9	3.8	3.8	3.8
Lactic acid, g/kg of DM		27.4	65.5	66.4	64.0
Acetic acid, g/kg of DM		22.9	12.9	9.7	10.2
Propionic acid, g/kg of DM		1.9	0.8	0.7	0.8
Butyric acid, g/kg of DM		0.3	0.4	0.3	0.3
Valeric acid, g/kg of DM		0.2	0.1	0.3	0.1
Ammonia nitrogen, g/kg of N		18.2	18.6	18.5	22.1

^1^ DM = dry matter, N = nitrogen.

**Table 3 animals-13-03042-t003:** Body weight, feed intake, and nutrient intake in dairy cows fed different broken rice substitutions in the diet.

	Broken Rice Substitution (%)					*p*-Value ^3^		
Items ^1^	0	12	24	36	SEM ^2^	L	Q	C
Body weight, kg	523.9	479.1	512.8	487.0				
Feed intake								
kg/day	13.6	13.4	12.7	12.6	1.29	0.51	0.96	0.83
% of BW	2.6	2.8	2.5	2.6	0.25	0.77	0.90	0.44
g/kg BW^0.75^	125.3	130.9	118.7	122.4	11.46	0.69	0.93	0.52
Nutrient intake (kg/day)								
Organic matter	12.8	12.7	12.0	12.0	1.23	0.61	0.99	0.83
Crude protein	2.5	2.2	2.0	1.8	0.19	0.02	0.71	0.87
Ether extract	0.8	0.7	0.6	0.6	0.06	0.01	0.93	0.88
Neutral detergent fiber	6.2	5.7	5.2	4.2	0.45	<0.01	0.65	0.74
Acid detergent fiber	3.2	3.1	2.6	2.2	0.23	0.04	0.47	0.63
Non-fiber carbohydrate	3.2	4.1	4.2	5.4	0.53	0.02	0.77	0.16

^1^ BW = body weight, BW^0.75^ = metabolic body weight. ^2^ SEM = standard error of the mean. ^3^ Orthogonal polynomial significant effects of broken rice substitutions (L = linear, Q = quadratic, C = cubic).

**Table 4 animals-13-03042-t004:** Apparent nutrient digestibility in dairy cows fed different broken rice substitutions in the diet.

	Broken Rice Substitution (%)					*p*-Value ^2^		
Digestibility, g/kg	0	12	24	36	SEM ^1^	L	Q	C
Dry matter	595	648	670	663	18.12	0.02	0.14	0.96
Organic matter	628	678	696	690	17.24	0.03	0.15	0.88
Crude protein	670	683	646	599	27.45	0.08	0.31	0.76
Ether extract	836	815	833	750	25.07	0.07	0.26	0.24
Neutral detergent fiber	467	579	596	606	38.93	0.04	0.24	0.62
Acid detergent fiber	354	403	383	293	39.58	0.31	0.12	1.00
Non-fiber carbohydrate	730	744	769	783	26.00	0.17	0.99	0.84

^1^ SEM = standard error of the mean. ^2^ Orthogonal polynomial significant effects of broken rice substitutions (L = linear, Q = quadratic, C = cubic).

**Table 5 animals-13-03042-t005:** Milk production and composition in dairy cows fed different broken rice substitutions in the diet.

	Broken Rice Substitution (%)					*p*-Value ^3^		
Items ^1^	0	12	24	36	SEM ^2^	L	Q	C
Milk production								
Milk yield, kg/day	16.1	16.1	14.7	14.8	2.27	0.62	0.98	0.78
FPCM, kg/day	14.2	13.7	13.5	12.8	1.82	0.60	0.97	0.93
ECM, kg/day	14.1	13.6	13.6	12.7	1.91	0.65	0.92	0.90
Protein, g/day	535.6	545.8	499.1	513.4	87.55	0.78	0.98	0.77
Fat, g/day	478.5	428.5	406.1	362.6	56.15	0.28	0.42	0.63
Lactose, g/day	753.8	768.8	713.0	703.5	125.47	0.72	0.92	0.84
Solid not fat, kg/day	1.4	1.4	1.3	1.3	0.23	0.76	0.98	0.88
Milk composition								
Milk protein, g/kg	33.4	33.6	33.9	33.8	1.90	0.87	0.92	0.97
Milk fat, g/kg	29.7	28.8	25.9	27.8	2.44	0.50	0.61	0.63
Milk lactose, g/kg	46.4	47.1	47.7	47.3	1.97	0.72	0.79	0.92
Solid not fat, g/kg	87.8	87.3	89.8	89.3	3.45	0.67	1.00	0.70
Milk energy, MJ/kg	2.3	2.3	2.3	2.4	0.10	0.66	0.40	0.68
SCC, ×10^3^ cell/mL	92.3	330.7	252.5	119.5	78.76	0.99	0.10	0.54

^1^ Fat and protein-corrected milk (FPCM), Energy-corrected milk (ECM), and milk energy (MJ/kg), SCC = somatic cell count. ^2^ SEM = standard error of the mean. ^3^ Orthogonal polynomial significant effects of broken rice substitutions (L = linear, Q = quadratic, C = cubic).

**Table 6 animals-13-03042-t006:** Daily respiratory gas (L/day), energy partition (kJ/kgBW^0.75^), intake (kJ/kgBW^0.75^), content of diet (MJ/kg of DM), and utilization in dairy cows fed different broken rice substitutions in the diets.

	Broken Rice Substitution (%)					*p*-Value ^3^		
Items ^1^	0	12	24	36	SEM ^2^	L	Q	C
Respiratory gas								
O_2_ consumption	3566	3787	4048	3849	471.71	0.63	0.68	0.82
CO_2_ production	4033	4360	4651	4491	538.40	0.53	0.67	0.87
CH_4_ emission	258.7	298.0	275.4	303.6	39.54	0.57	0.89	0.54
RQ	1.1	1.2	1.2	1.2	0.03	0.69	0.93	0.91
Energy partition								
Gross energy intake	2190	2368	2066	2411	149.80	0.63	0.60	0.12
Fecal excretion	597.0	627.7	561.4	620.6	59.51	0.99	0.82	0.42
Urine excretion	55.3	60.1	56.5	44.7	4.50	0.09	0.08	0.94
Methane emission	92.2	116.0	100.2	115.7	13.10	0.40	0.77	0.26
Heat production	692.1	799.5	811.4	796.0	90.10	0.43	0.56	0.85
Milk energy	287.4	352.3	319.5	323.0	51.44	0.61	0.69	0.53
Energy balance	466.4	411.9	416.9	472.0	149.07	0.87	0.38	0.45
E_l(0)_	638.1	652.1	519.9	670.0	123.13	0.95	0.55	0.43
Energy intake								
DE	1593	1739	1505	1753	125.39	0.57	0.60	0.14
ME	1446	1564	1348	1591	133.79	0.61	0.56	0.19
NE_L_	1151	1178	1023	1185	167.21	0.98	0.66	0.52
Energy content								
DE	13.1	13.4	13.2	13.3	0.48	0.84	0.91	0.78
ME	11.9	12.0	11.8	12.1	0.57	0.84	0.86	0.74
NE_L_	9.4	9.0	9.0	9.1	1.14	0.88	0.86	0.99
Energy utilization								
DE/GE	0.72	0.74	0.73	0.74	0.02	0.60	0.95	0.69
ME/GE	0.65	0.66	0.65	0.68	0.03	0.65	0.74	0.68
ME/DE	0.90	0.90	0.89	0.91	0.02	0.83	0.48	0.67

^1^ E_l(0)_ = milk energy output adjusted to zero energy balance; DE = digestible energy; GE = gross energy, ME = metabolizable energy; NE_L_ = net energy for lactation. ^2^ SEM = standard error of the mean. ^3^ Orthogonal polynomial significant effects of broken rice substitutions (L = linear, Q = quadratic, C = cubic).

**Table 7 animals-13-03042-t007:** The regression equations used for estimating energy values in broken rice.

Item ^1^	Regression Equations ^2^	RMSE ^3^	R^2^	*p*-Value	Slope		Intercept	
					SEM ^4^	*p*-Value	SEM	*p*-Value
DE, MJ/kg DM	*Y* = 13.13*X* + 0.02	0.25	0.98	<0.0001	0.49	<0.0001	0.10	0.85
ME, MJ/kg DM	*Y* = 11.87*X* + 1.91	0.31	0.96	<0.0001	0.62	<0.0001	0.13	0.89
NE_L_, MJ/kg DM	*Y* = 8.68*X* + 1.18	0.50	0.85	<0.0001	0.99	<0.0001	0.21	0.95

^1^ DE = digestible energy, DM = dry matter, ME = metabolizable energy, and NE_L_ = net energy for lactation. ^2^
*Y* is ingredient-associated energy for lactation in megajoules, *X* is test ingredient intake in kilograms (DM basis), the intercept is in megajoules, and the slopes are in megajoules per kilogram DM. ^3^ RMSE = root-mean square error. ^4^ SEM = standard error of the mean.

**Table 8 animals-13-03042-t008:** Comparison between substitution and the regression method of broken rice energy content.

Item ^1^	Comparative Method				SEM ^2^	*p*-Value
	Substitution	Substitution	Substitution	Regression		
	12%	24%	36%			
DE, MJ/kg DM	13.15	13.13	12.82	13.13	0.63	0.98
ME, MJ/kg DM	11.88	11.87	11.60	11.87	0.77	0.99
NE_L_, MJ/kg DM	8.61	8.72	8.45	8.68	1.20	0.99

^1^ DE = digestible energy, DM = dry matter, ME = metabolizable energy, and NE_L_ = net energy for lactation. ^2^ SEM = standard error of the mean.

## Data Availability

Data are available upon reasonable request.

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
