# Peer review of "The Energy Contents of Broken Rice for Lactating Dairy Cows"

_animals, 2023, doi:10.3390/ani13193042_

Round 1
Reviewer 1 Report (Previous Reviewer 2)
Corrections made are satisfactory for publication
Author Response
Response to Reviewer 1 Comments
Comments and Suggestions for Authors
Corrections made are satisfactory for publication.
Response: We thank you for your insight reviewing and valuable comments that substantially helped us improve the manuscript.
Reviewer 2 Report (Previous Reviewer 3)
After resubmition I have gone through the article and it has improved exponentially. The article now seems good and is now ready for publication. Only minor spelling check needed.
Only minor spelling check needed.
Author Response
Response to Reviewer 2 Comments
Comments and Suggestions for Authors
After resubmission, I have gone through the article, and it has improved exponentially. The article now seems good and is now ready for publication. Only a minor spelling check is needed.
Response: We thank you for your insight reviewing and valuable comments that substantially helped us improve the manuscript.
Reviewer 3 Report (New Reviewer)
Evaluating the energy contents of broken rice for lactating dairy cows using animal models in a respiratory chamber is somewhat a big research endeavour. However, the valididty of the results is dependent upon appropriate methodology although statistical models can be used to predict net energy of lactation based on chemical analysis results, in vivo models remain the gold standard. The writing of the manuscript is very good but fundamental issues have been identified in the methodology.
L93: change fermented feed to silage.
L112: 5000 kg capacity () mixer.
Diet: the diet is somewhat problematic. The authors fed a total mixed ration and the only major fibre source is Rice straw at 15-9.6% inclusion level. The effect on rumination and overall animal response is faulty.
Raw cassava pulp itself will contain significantly high soluble carbohydrates although high fibre. Howowever, the only major fibre source is rice straw and at 15% inclusion, having an average NDF and ADF content of 46% and 23.5% is very outrageos. Was the values reported in Table 2 based on calculated nutrient content or analysed composition? It is hard to juxtapose that 15% inclusion of rice straw, 12% rice, 11% rice bran and 30% cassava pulp, among others, will give an NDF of >45%. By my estimation, the expected NDF from the experimental diet will be too low for Dairy cows.
-Calibration of gas analyser relied on two oxygen concentrations of 19% and 20.6%. The data points will be too close to achieve a very reliable straight-line equation.
Statistical analysis
-The model statement and its analysis is somewhat fraght with some error. It is not appropriate to consider effect of period as a fixed effect. With a single 4X 4 latin square, the results obtained are usually very doubtful because of potential carry-over effect and limited number of animal use. More appropriate is the repeated latin square/double latin square etc. In all cases, period effect should be a random effect.
Result
-Authors need to provide more details on the nature of ill health experienced by the animal.
Discussion
The authors observed significant increase in non-fibre carbohydrate intake (386-387), obviously because the broken rice provided more of soluble than rumen effective fibre thus validating the suspicion that the diets were formulated without adequate consideration for fibre.
While the study objectives and rationale are clearly defined, several methodology issues need to be addressed or justified.
The English language quality is adequate. However, final editing work is necessary.
Author Response
Response to Reviewer 3 Comments
Comments and Suggestions for Authors
Evaluating the energy contents of broken rice for lactating dairy cows using animal models in a respiratory chamber is somewhat of a big research endeavor. However, the validity of the results is dependent upon appropriate methodology although statistical models can be used to predict the net energy of lactation based on chemical analysis results, in vivo models remain the gold standard. The writing of the manuscript is very good but fundamental issues have been identified in the methodology.
Point 1:
L93: change fermented feed to silage.
Response 1: We make changes accordingly.
Point 2:
L112: 5000 kg capacity () mixer.
Response 2: We make changes accordingly.
Point 3:
Diet: the diet is somewhat problematic. The authors fed a total mixed ration and the only major fibre source is Rice straw at 15-9.6% inclusion level. The effect on rumination and overall animal response is faulty.
Response 3: Thank you for your observation. To improve clarity, we added our discussion in the Discussion section (L373-390) as follows.
“In this study, dietary NDF analyzed ranges from 33.7% to 45.8% indicating that the diets provided sufficient dietary NDF fiber requirement for dairy cows [39]. Our previous work indicates that 10% of rice straw included in the total mixed ration of native Thai cattle and Holstein crossbred bulls maintained an average daily 8.5 to 9.85 h chewing time and 3.1 to 4.7 h ruminating time suggesting that most tropical feed and systems depend on agricultural industry by-products such as rice straw, cassava pulp, and palm kernel cake that have a high NDF and indigestible NDF fraction that may be contributing to the rumen floating mat and stimulates sufficient ruminating activities (2.5 to 10.5 h/d). Most tropical feed and systems depend on agricultural industry by-products such as rice straw, cassava pulp, and palm kernel cake that have a high NDF and indigestible NDF fraction that may be stimulates chewing activities and contributes to the rumen floating mat. Analyzed chemical analysis (Table 2) dietary NDF ranges from 33.7% to 45.8% with 21.5% to 22.8% total forage NDF concentration indicating that the diets provided sufficient dietary NDF fiber requirement for dairy cows [39]. NRC [39] recommended minimum concentrations of total dietary NDF for cows at 25% with the condition that 19% of dietary NDF is from forage. In this study, rice straw is mainly the total forage NDF concentration contribution calculated ranged from 21.5% to 22.8%, therefore, it may be expected that the physical structure of forage could maintain chewing, ruminal, and cow health.”
References:
1.Gunha, T.; Kongphitee, K.; Sommart, K. 2016. Feed intake, digestibility, growth performances and eating behavior of native Thai beef cattle fed diets differing in energy density using cassava pulp with rice straw. 1st International Conference on Tropical Animal Science and Production (TASP 2016), Ambassador hotel, Thailand, Date of conference (26-29 July 2016).
- Binsulong, B., Gunha, T, Kongphitee, K., Maeda, K. and K. Sommart. 2023. Enteric Methane Emissions, Rumen Fermentation Characteristics, and Energetic Efficiency of Holstein Crossbred Bulls Fed Total Mixed Ration Silage” for consideration for publication as an article in Fermentation. (submitted)
- National Research Council (NRC). Nutrient requirement of dairy cattle. 7th ed.; National Academic Press: Washington, DC, USA, 2001.
Point 4:
Raw cassava pulp itself will contain significantly high soluble carbohydrates although high fibre. Howowever, the only major fibre source is rice straw and at 15% inclusion, having an average NDF and ADF content of 46% and 23.5% is very outrageos. Was the values reported in Table 2 based on calculated nutrient content or analysed composition? It is hard to juxtapose that 15% inclusion of rice straw, 12% rice, 11% rice bran and 30% cassava pulp, among others, will give an NDF of >45%. By my estimation, the expected NDF from the experimental diet will be too low for Dairy cows.
Response 4: Thank you for your observation. To improve clarity, we rewrote the passage and added details like in Point 3 in the Discussion section (L373-390) as follows.
“In this study, dietary NDF analyzed ranges from 33.7% to 45.8% (see Analyzed and Calculated in Table 1, 2 below) indicating that the diets provided sufficient dietary NDF fiber requirement for dairy cows [39]. Our previous work indicates that 10% of rice straw included in the total mixed ration of native Thai cattle and Holstein crossbred bulls maintained an average daily 8.5 to 9.85 h chewing time and 3.1 to 4.7 h ruminating time suggesting that most tropical feed and systems depend on agricultural industry by-products such as rice straw, cassava pulp, and palm kernel cake that have a high NDF and indigestible NDF fraction that may be contributing to the rumen floating mat and stimulates sufficient ruminating activities (2.5 to 10.5 h/d). Most tropical feed and systems depend on agricultural industry by-products such as rice straw, cassava pulp, and palm kernel cake that have a high NDF and indigestible NDF fraction that may be stimulates chewing activities and contributes to the rumen floating mat. Analyzed chemical analysis (Table 2) dietary NDF ranges from 33.7% to 45.8% with 21.5% to 22.8% total forage NDF concentration indicating that the diets provided sufficient dietary NDF fiber requirement for dairy cows [39]. NRC [39] recommended minimum concentrations of total dietary NDF for cows at 25% with the condition that 19% of dietary NDF is from forage. In this study, rice straw is mainly the total forage NDF concentration contribution calculated ranged from 21.5% to 22.8%, therefore, it may be expected that the physical structure of forage could maintain chewing, ruminal, and cow health.”
References
Table 1. Analyzed chemical composition and fermentation quality of broken rice and the four broken rice substitution dietary treatments.
|
Broken rice substitution (%) |
|||||
|
Items1 |
Broken rice |
0 |
12 |
24 |
36 |
|
Chemical composition |
|||||
|
Dry matter, % |
88.9 |
32.7 |
34.0 |
36.1 |
40.8 |
|
Organic matter, %DM |
99.2 |
93.6 |
94.5 |
95.1 |
95.4 |
|
Crude protein, %DM |
7.4 |
18.5 |
16.7 |
15.5 |
14.5 |
|
Neutral detergent fiber, %DM |
6.4 |
45.8 |
42.2 |
41.3 |
33.7 |
|
Acid detergent fiber, %DM |
1.2 |
23.5 |
23.2 |
20.5 |
17.1 |
|
Ether extract, %DM |
1.3 |
5.9 |
5.3 |
5.1 |
4.5 |
|
Non-fiber carbohydrate, %DM |
84.9 |
23.5 |
30.3 |
33.1 |
42.7 |
1 DM = dry matter, non-fiber carbohydrate calculated as 100 − (%NDF + %CP + %EE + %ash), N = nitrogen.
Table 2 Calculated dietary total NDF and forage NDF content.
|
Items |
NDF |
%NDF fraction of dietary treatments |
|||
|
%DM |
0 |
12 |
24 |
36 |
|
|
Feedstuffs |
|
|
|
|
|
|
Rice straw |
68.0 |
10.2 |
9.0 |
7.8 |
6.5 |
|
Broken rice |
4.3 |
0.0 |
0.5 |
1.0 |
1.5 |
|
Cassava pulp |
43.0 |
12.9 |
11.3 |
9.8 |
8.2 |
|
Wet brewery waste |
74.7 |
11.2 |
9.9 |
8.5 |
7.2 |
|
Rice bran |
24.2 |
2.7 |
2.3 |
2.0 |
1.7 |
|
Palm kernel cake |
33.5 |
4.0 |
3.5 |
3.1 |
2.6 |
|
Soybean meal |
25.7 |
3.9 |
3.4 |
2.9 |
2.5 |
|
Urea |
- |
0.0 |
0.0 |
0.0 |
0.0 |
|
Mineral |
- |
0.0 |
0.0 |
0.0 |
0.0 |
|
Premix |
- |
0.0 |
0.0 |
0.0 |
0.0 |
|
Total NDF concentration (%) |
- |
44.8 |
39.9 |
35.1 |
30.2 |
|
Forage NDF concentration (%) |
- |
22.8 |
22.5 |
22.2 |
21.5 |
- National Research Council (NRC). Nutrient requirement of dairy cattle. 7th ed.; National Academic Press: Washington, DC, USA, 2001.
Point 5:
-Calibration of gas analyser relied on two oxygen concentrations of 19% and 20.6%. The data points will be too close to achieve a very reliable straight-line equation.
Response 5: Thank you for your observation. We agree on our missing and revised a correction to the missing two oxygen concentrations data from “19% and 20.6%” to “18.90% and 24.96%”.
Point 6:
Statistical analysis:
The model statement and its analysis is somewhat fraght with some error. It is not appropriate to consider effect of period as a fixed effect. With a single 4X 4 latin square, the results obtained are usually very doubtful because of potential carry-over effect and limited number of animal use. More appropriate is the repeated latin square/double latin square etc. In all cases, period effect should be a random effect.
Response 6: Thank you for your recommendation. To improve clarity, we rewrote it as follows:
Materials and Methods (L233-239) we rewrote it as follows:
“All experimental data were subject to analysis of variance using the general linear model of SAS [29] for a 4 × 4 Latin square design using the model:
Yijk = μ + ρi + γj + τk+ εijk,
where Yijk was the response of the dependent variable, μ was the observation means, ρi was the random effect of the period (i = 1 to 4), γj is the random effect of the cow (j = 1 to 4), τk is the fixed effect of dietary treatment (k = 1 to 4), and εijk is the residual error. Orthogonal polynomial analysis was conducted to determine a linear, quadratic, and cubic effect of dietary treatment.”
Discussion section (L441-445) In addition, we have added limitations of the study using a 4x4 Latin square for clarity in the and as follows.
“Although the advantage of a 4x4 Latin square design is allowed to determine the effect of cow, period, and the change responses to dietary treatment. However, the short-term measurement (21 d) and the limited number of cows per treatment are limitations of this study. Dairy cow production performance may be required to be confirmed in the long-term feeding experiment with more animals.”
Conclusion (L461-462) To improve clarity, we rewrote it as follows:
“Long-term feeding experiments are needed to develop dairy cattle-fed broken rice feeding systems.”
Point 7:
Result: Authors need to provide more details on the nature of ill health experienced by the animal.
Response 7: Thank you for your suggestion. To improve clarity, we rewrote the passage and added details in the Results section (L243-248) as follows.
“Fifteen of a possible 16 energy balances were completed. During the data collection of the third period, one cow was removed from total tract digestibility and respiratory gas collections became ill of hardware disease. The consulted veterinarian treated hardware disease including antibiotics and administered a magnet into the rumen and the cow was used for data collection in the fourth period after feed intake and milk yield recovered well by 12 days. “
Point 8:
Discussion:
The authors observed significant increase in non-fibre carbohydrate intake (386-387), obviously because the broken rice provided more of soluble than rumen effective fibre thus validating the suspicion that the diets were formulated without adequate consideration for fibre.
Response 8: Thank you for your recommendation. To improve clarity, we rewrote the passage and added details like in Point 3 in the Discussion section (L373-390) as follows.
“In this study, dietary NDF analyzed ranges from 33.7% to 45.8% indicating that the diets provided sufficient dietary NDF fiber requirement for dairy cows [39]. Our previous work indicates that 10% of rice straw included in the total mixed ration of native Thai cattle and Holstein crossbred bulls maintained an average daily 8.5 to 9.85 h chewing time and 3.1 to 4.7 h ruminating time suggesting that most tropical feed and systems depend on agricultural industry by-products such as rice straw, cassava pulp, and palm kernel cake that have a high NDF and indigestible NDF fraction that may be contributing to the rumen floating mat and stimulates sufficient ruminating activities (2.5 to 10.5 h/d). Most tropical feed and systems depend on agricultural industry by-products such as rice straw, cassava pulp, and palm kernel cake that have a high NDF and indigestible NDF fraction that may be stimulates chewing activities and contributes to the rumen floating mat. Analyzed chemical analysis (Table 2) dietary NDF ranges from 33.7% to 45.8% with 21.5% to 22.8% total forage NDF concentration indicating that the diets provided sufficient dietary NDF fiber requirement for dairy cows [39]. NRC [39] recommended minimum concentrations of total dietary NDF for cows at 25% with the condition that 19% of dietary NDF is from forage. In this study, rice straw is mainly the total forage NDF concentration contribution calculated ranged from 21.5% to 22.8%, therefore, it may be expected that the physical structure of forage could maintain chewing, ruminal, and cow health.”
Point 8:
While the study objectives and rationale are clearly defined, several methodological issues need to be addressed or justified.
Response 8: Thank you for your helpful feedback. To improve clarity, we rewrote the passage and added details in the Methos and Discussion section accordingly.
Round 2
Reviewer 3 Report (New Reviewer)
The authors have significantly improved the manuscript by attending to most of the questions in the previous review. A few more corrections are here recommended:
L374: change substitution to substituting.
L378-388: Some statements are repeated. Authors need to re-read and correct.
L392: rice straw is the main source of dietary NDF…
The authors did not respond to the earlier comments in form of a rebuttal letter. How the NDF value of the control diet amount to 45.8% when indeed Rice straw (main NDF souce) was included at 15%.
Nutrient level is: nutrient composition X inclusion level.
If this is applied for Rice straw, Cassava pulp, Rice bran, PKC, soybean meal and Wet brewery waste, the total NDF is 45.8. This is the authors’ claims. I have doubts this will be correct.
On the statistical analysis
The authors have re-written the relevant section of the statistical analysis to show that Period effect is a random (not fixed effect). However, there is no evidence that the statistical analysis was carried out afresh using the corrected statistical model to arrive at the data presented. This needs to be done.
On Feed formulation of experimental Diet. The authors presented the following values to estimate the animal diet:
Cassava pulp with NDF of 43%
Wet brewery waste NDF= 74.7%
Soybean meal NDF = 25.7%
I find these values quite outrageous and therefore very doubtful if the true NDF in the control diet was 45.8%
There are only very minor language issues with the manuscript, some of which can be corrected during typesetting.
Author Response
Point-by-point Responses to Reviewer Comments
Manuscript ID: animals-2535231
Type of manuscript: Article
Title: The energy contents of broken rice for lactating dairy cows
Authors: Thidarat Gunha, Kanokwan Kongphitee, Bhoowadol Binsulong, Kritapon Sommart *
Response to Reviewer 3 Comments
The authors have significantly improved the manuscript by attending to most of the questions in the previous review. A few more corrections are here recommended:
Point 1:
L374: change substitution to substituting.
Response 1: We make changes accordingly.
Point 2:
L378-388: Some statements are repeated. Authors need to re-read and correct.
Response 2: Thank you for your observation. We agree on our missing and revised a correction.
Point 3:
L392: rice straw is the main source of dietary NDF. The authors did not respond to the earlier comments in form of a rebuttal letter. How the NDF value of the control diet amount to 45.8% when indeed Rice straw (main NDF source) was included at 15%. The nutrient level is: nutrient composition X inclusion level. If this is applied for Rice straw, Cassava pulp, Rice bran, PKC, soybean meal and Wet brewery waste, the total NDF is 45.8. This is the authors’ claims. I have doubts this will be correct.
Response 3: Yes, correct. For example (Table 1), the total dietary NDF of 0% broken rice treatment contained NDF = 43.7% was a summation of NDF from rice straw (10.3%), cassava pulp (11.7%), brewery waste (8.3%), rice bran (2.2%), palm cake (9.0%) and soybean meal (2.1%).
Table 1 Calculated total dietary NDF and forage NDF content.
|
Items |
NDF1 |
%NDF fraction of dietary |
|||
|
%DM |
0% |
12 |
24 |
36 |
|
|
Feedstuff |
|
|
|
|
|
|
Rice straw |
68.5 |
10.3 |
9.0 |
7.8 |
6.6 |
|
Broken rice |
4.1 |
0.0 |
0.5 |
1.0 |
1.4 |
|
Cassava pulp |
38.9 |
11.7 |
10.3 |
8.9 |
7.5 |
|
Wet brewery waste |
55.3 |
8.3 |
7.3 |
6.3 |
5.3 |
|
Rice bran |
20.3 |
2.2 |
2.0 |
1.7 |
1.4 |
|
Palm kernel cake |
75.3 |
9.0 |
7.9 |
6.9 |
5.8 |
|
Soybean meal |
14.3 |
2.1 |
1.9 |
1.6 |
1.4 |
|
Urea |
- |
- |
- |
- |
- |
|
Mineral |
- |
- |
- |
- |
- |
|
Premix |
- |
- |
- |
- |
- |
|
Total dietary NDF, % (calculated) |
- |
43.7 |
38.9 |
34.1 |
29.4 |
|
Forage NDF, % |
- |
23.5 |
23.3 |
22.9 |
22.4 |
1 Working Committee of Thai Feeding Standard for Ruminants (WTSR). Nutrient Requirements of Dairy Cattle in Thailand. 1st ed.; Khon Kaen University Press: Khon Kaen, Thailand, 2021.
Point 4:
On Feed formulation of experimental Diet. The authors presented the following values to estimate the animal diet:
Cassava pulp with NDF of 43%
Wet brewery waste NDF= 74.7%
Soybean meal NDF = 25.7%
I find these values quite outrageous and therefore very doubtful if the true NDF in the control diet was 45.8%
Response 4: Thank you for your observation. We corrected the feedstuffs NDF database (WTSR, 2021) as in Table 2. The calculated %NDF now ranges from 29.4 to 43.7.
Table 2 Calculated total dietary NDF and forage NDF content.
|
Items |
NDF1 |
%NDF fraction of dietary |
|||
|
%DM |
0% |
12 |
24 |
36 |
|
|
Feedstuff |
|
|
|
|
|
|
Rice straw |
68.5 |
10.3 |
9.0 |
7.8 |
6.6 |
|
Broken rice |
4.1 |
0.0 |
0.5 |
1.0 |
1.4 |
|
Cassava pulp |
38.9 |
11.7 |
10.3 |
8.9 |
7.5 |
|
Wet brewery waste |
55.3 |
8.3 |
7.3 |
6.3 |
5.3 |
|
Rice bran |
20.3 |
2.2 |
2.0 |
1.7 |
1.4 |
|
Palm kernel cake |
75.3 |
9.0 |
7.9 |
6.9 |
5.8 |
|
Soybean meal |
14.3 |
2.1 |
1.9 |
1.6 |
1.4 |
|
Urea |
- |
- |
- |
- |
- |
|
Mineral |
- |
- |
- |
- |
- |
|
Premix |
- |
- |
- |
- |
- |
|
Total dietary NDF, % (calculated) |
- |
43.7 |
38.9 |
34.1 |
29.4 |
|
Forage NDF, % (calculated) |
- |
23.5 |
23.3 |
22.9 |
22.4 |
1 Working Committee of Thai Feeding Standard for Ruminants (WTSR). Nutrient Requirements of Dairy Cattle in Thailand. 1st ed.; Khon Kaen University Press: Khon Kaen, Thailand, 2021.
Point 5:
On the statistical analysis
The authors have re-written the relevant section of the statistical analysis to show that the period effect is a random (not fixed effect). However, there is no evidence that the statistical analysis was carried out afresh using the corrected statistical model to arrive at the data presented. This needs to be done.
Response 5: Thank you for your observation. We agreed on our missing carried out statistical analysis. We revised the data change of SEM and p-value in Table 3 – 6.
We also make corrections in the result, discussion, and conclusion, abstract section (L 249-255, L269-270, L388-394, L437-441, L27-31).
Point 6:
There are only very minor language issues with the manuscript, some of which can be corrected during typesetting.
Response 6: We thank you for your insight reviewing and valuable comments that substantially helped us improve the manuscript.

Round 3
Reviewer 3 Report (New Reviewer)
The authors have corrected all the relevant section
The earlier comments on language suffices. Only very minor language issues need correcting.
Author Response
Response to Reviewer 3 Comments
The authors have corrected all the relevant sections.
The earlier comments on language suffice. Only very minor language issues need correcting.
Response 6: We make corrected changes accordingly. We thank you for your insight reviewing and valuable comments that substantially helped us improve the manuscript.

This manuscript is a resubmission of an earlier submission. The following is a list of the peer review reports and author responses from that submission.
Round 1
Reviewer 1 Report
I consider the article to have serious flaws that led me to suggest rejection.
1. Only 01 Latin Square (04 cows x four periods x four diets)
There are no residual degrees of freedom in this case to support the statistical analysis:
Total: 15
cow = 3
Period = 3
Diet = 3
Residual = 06.0 which is very, very low to lend credibility to the statistical analysis.
Another problem, the objective was to evaluate the energy contents of broken rice, but the diets will be completely different, with levels of
Cassava pulp (30.0 26.4 22.8 19.2 % of dry matter), Wet brewery waste (15.0 13.2 11.4 9.6, % of dry matter), Rice bran (11.0 9.7 8.4 7.0, % of dry matter) , Palm kernel cake (12.0 10.5 9.1 7.7, % of dry matter) and Soybean meal 15.0 13.2 11.4 9.6, % of dry matter).
How to evaluate energy contents of broken rice with a variation in ingredients of this magnitude???
Author Response
Response to Reviewer 1 Comments
I consider the article to have serious flaws that led me to suggest rejection.
Point 1: 1. Only 01 Latin Square (04 cows x four periods x four diets). There are no residual degrees of freedom in this case to support the statistical analysis:
Total: 15 cow = 3 Period = 3 Diet = 3 Residual = 06.0 which is very, very low to lend credibility to the statistical analysis.
Response 1: Latin square designs are often used in dairy cattle nutrition experiments where subjects are allocated treatments over a given period, and time is thought to have a major effect on the experimental response. Reducing the experimental error by accounting for column and row variability is possible. If more than three blocks and treatments exist, then Latin square designs are possible. The Latin square also provides better efficiency than the RCBD in supporting the principle of 3 Rs (Reduction, Refinement, Replacement) for animal research ethical issues.
The advantage of the Latin square design and statistical analysis is to control the variation from different blocks and experimental runs. This study assigns treatments to blocks differently in the Latin square design, represented as columns (animals) and rows (periods). Each column and each row are a complete block of all treatments. In our study, a 4x4 Latin square, three explained sources of variability are defined: animal, period, and treatment. Treatment is assigned just once in each row and column. Each animal will receive all treatment in different periods. In that sense, the Latin square is a changeover design. Our results indicate a sufficient coefficient of variation (e.g. Dry matter intake of dairy cows; CV= 7.7%; see Table 3). Recent research using a 4x4 Latin squire design in dairy cows was published (e.g. doi.org/10.1016/j.anifeedsci.2022.115454 ).
Point 2: Another problem, the objective was to evaluate the energy contents of broken rice, but the diets will be completely different, with levels of Cassava pulp (30.0 26.4 22.8 19.2 % of dry matter), Wet brewery waste (15.0 13.2 11.4 9.6, % of dry matter), Rice bran (11.0 9.7 8.4 7.0, % of dry matter) , Palm kernel cake (12.0 10.5 9.1 7.7, % of dry matter) and Soybean meal 15.0 13.2 11.4 9.6, % of dry matter). How to evaluate the energy contents of broken rice with a variation in ingredients of this magnitude???
Response 2: “Substitution and regression methods are required for evaluating the energy content of individual feedstuffs, and a regression method based on multiple-point substitution by regress is more accurate. Therefore, this study involves different rations of broken substitution into basal diets.” The previous study has summarized methods of energy evaluation of feed ingredients and their accuracy (Wei et al., 2018 (DOI:10.1080/1745039X.2018.1482076); Adelola et al., 2019 (doi:10.3382/ps.2008-00187)).
For more information, we added this description in Section 2.1 (Lines 102-108). The text now reads, “Substitution and regression methods are required for evaluating the energy content of individual feedstuffs, and a regression method based on multiple-point substitution. Therefore, this study involves different rations of broken substitution into basal diets.”
The substitution and regression methods were used to determine diets' metabolizable and net energy values, according to Wei et al., 2018 (DOI:10.1080/1745039X.2018.1482076). The calculation is described in detail in section 2.5.
Our data indicated that the regression equation for the DE, ME, and NE was significantly precision (R2 = 0.85-0.98, RMSE = 0.25-0.50, p < 0.0001) and accurate within the previous work (see Section 3.6, 4.3).
Reviewer 2 Report
General consideration: The general purpose of the article was to determine the energy contest of broken rice in dairy cows and the effects of various levels of broken rise on dairy cow performance.
The introduction should be improved by reporting more bibliographic references on the use of broken rice in cattle. The materials and methods section lacks information that is then discussed in the results section.
2. Materials and Methods
2.2 Feed intake and digestibility
L. 137 Missing information on how feces and urine were measured.
L. 139 Data are not given in any of the succeeding tables.
2.4 Sample Collection and Chemical Analysis
Lack of information on how correct milk was calculated and how Lactic and acetic acids were measured.
4. Discussion
L. 354 DM and OM intake o measured? If you have measured how the measurement occurred and the data must be reported in the table
L377 correct with....estimated digestibility
Author Response
Response to Reviewer 2 Comments
General consideration: The general purpose of the article was to determine the energy content of broken rice in dairy cows and the effects of various levels of broken rice on dairy cow performance.
Point 1: The introduction should be improved by reporting more bibliographic references on the use of broken rice in cattle. The materials and methods section lacks information that is then discussed in the results section.
Response 1: Thank you for your suggestion. We made changes accordingly.
- Materials and Methods
2.2 Feed Intake and Digestibility
Point 2: L. 137 Missing information on how feces and urine were measured.
Response 2: We described details in L138-147.
Point 3: L. 139 Data are not given in any of the succeeding tables.
Response 3: We rearranged Table 2 into Sec 3.1 accordingly.
2.4 Sample Collection and Chemical Analysis
Point 4: Lack of information on how correct milk was calculated and how Lactic and acetic acids were measured.
Response 4: Thank you for your query. The text now reads, “Fat and protein-corrected milk (kg/day) = (0.337 + 0.116 × fat% + 0.06 × protein%) × milk yield (kg/day), Energy-corrected milk (kg/day) = ((milk yield, kg/day) × milk energy (MJ/kg))/3.1. Feed efficiency = FPCM yield (kg/day)/DMI (kg/day). Milk energy (MJ/kg) = (0.0384 × fat%) + (0.0223 × protein%) + (0.0199 × lactose%) − 0.108.”
- Discussion
Point 5: L. 354 DM and OM intake o measured? If you have measured how the measurement occurred and the data must be reported in the table
Response 5: We reported the dry and organic matter intake data in Table 3.
Point 6: L377 correct with....estimated digestibility
Response 6: We made changes accordingly.
Reviewer 3 Report
The current submitted manuscript by Thidarat Gunha et al. is a very good manuscript on the topic of ‘The Energy content of broken rice for lactating dairy cows. I have gone through the complete manuscript and the following points should be taken as suggestion.
1. Broken rice use on the diet of animal production is a good option to reduce the feed cost in the tropical areas.
2. Use of urea in the feed may be discouraged for organic milk production.
3. Molasses/ Jaggery and dicalcium phosphate may be added to improve the protein level.
4. While testing the fermented feed data may be taken to identify any antimetabolic effect is there or not.
The english language is fine only need some minor spelling and grammer corrections.
Author Response
Response to Reviewer 3 Comments
Comments and Suggestions for Authors
The current submitted manuscript by Thidarat Gunha et al. is a very good manuscript on the topic of ‘The Energy content of broken rice for lactating dairy cows. I have gone through the complete manuscript and the following points should be taken as suggestion.
Point 1: 1. Broken rice use on the diet of animal production is a good option to reduce the feed cost in the tropical areas.
Response 1: Thank you for your suggestion. We added this recommendation in the Discussion section 4.1
Point 2: 2. Use of urea in the feed may be discouraged for organic milk production.
Response 2: We agree to our result and application limitation for organic milk production.
Point 3: 3. Molasses/ Jaggery and dicalcium phosphate may be added to improve the protein level.
Response 3: We agree to our result and application limitation for organic milk production.
Point 4: 4. While testing the fermented feed data may be taken to identify any antimetabolic effect is there or not.
Response 4: Based on our chemical analysis and previous report, as in Discussion section 4.1, our data indicated that the fermentation qualities were characterized by low pH (< 3.9) and high lactic acid content after seven days of ensiling, indicating that all the diets were good-quality silages, like a previous report by Kotupan and Sommart [7], Wang and Nishino [28] and Kongphitee et al. [29].
Comments on the Quality of English Language
Point 5: The english language is fine only need some minor spelling and grammer corrections.
Response 5: Thank you for your suggestion. We made changes in the English language accordingly.
Reviewer 4 Report
animals-2393384 "The energy contents of broken rice for lactating dairy cows"
This is a comprehensive and useful experiment. The information provided is generally sound and correct, although further details are required. The writing is generally excellent and the data is mostly discussed appropriately. Some changes can be made to improve the manuscript prior to publication. The information is highly relevant to the local and potentially international industries.
General comments:
66-68 – instead of repeating 30.9% perhaps reword to “replacing steam-flaked corn with steam-flaked brown rice (30.9%) did not alter milk yield but increased milk fat production” or similar
70-79 – while the information in monogastric species is interesting it is not very relevant to ruminants. Suggest condensing this section.
91-95 – what parity were the cows? Were they inseminated at any stage? Were cows milked in their pens or walked to a milking parlour? If walked, please describe distance etc.
96-100 – were all cows on the same diet within each period or was it one cow per diet per period? Please clarify.
Table 1 – be consistent with the number of significant figures reported in the footnotes (minerals).
128-134 – you weighed residual feed but it is not possible to accurately determine which portion of the diet was consumed given it is a TMR. This needs to be discussed as an experimental flaw.
135-144 – describe the met cages in more detail
146-165 – it is not possible from the description here or in the cited paper (Suzuki et al 2008) to adequately understand what the chambers / met cages were like. Were the animals able to lay down? Or were their heads in the ‘hood’ section for the whole period. What ambient temperature were the animals kept at? Were they indoors? What bedding (if any) was used? How was food delivered (in feeders, or on the floor?). was the feed in the hood area? Was the water also present in the hood area? If the animals were able to move their heads out of the hood, how many hrs a day were they actually sampled for?
168 – do you mean “fan forced” oven? “until” reaching a consistent weight?
180 – do you mean fresh as in just mixed or fresh as in +7days after mixing?
Table 3 – add hyphen in “nonfiber”
Table 3, 4 & 5 – please clarify in the title that n=166 is n=4 per treatment group and not n=16 per group as the current formatting implies.
Tables – check formatting. It may be a PDF issue but it looks like some ‘tick marks’ or small borders are showing at the top and bottom of the tables.
Table 5 – the “1” superscript is listed twice in the table. Please correct.
Table 6 – n = 15? Was one cow removed? Why? This needs to be described (this is also not mentioned later when n=15 data is presented). The results section should start with a statement as to why data from one animal was removed/not recorded etc. Also, update to be clear about n per group as described previously.
289 – “Methane energy lost cubically” due to what? Increasing BR content? Please clarify.
Inconsistent use of BR abbreviation throughout, please be consistent.
Figure 1 – spacing/formatting of the data point symbols in the caption are not quite right (brackets partially covering the symbols). Please check.
Table 7. Formatting – why is DE bold and underlined? Incorrect spelling “slope”, superscript 1 missing (not in the table but in the footnote),
Table 8 – again why is DE bold and underlined?
356-359 – this sentence is confusing and I cannot follow.
378-380 – “Kendall et al. [31] indicated that the NDF intake was higher for cows fed 28% compared to 23% NDF in the diet, resulting in an increase in milk fat production.” – yes I would assume NDF intake was higher when cows were fed diets with higher NDF. I suggest rewordinig.
365-380 – Did you measure the PUFA levels in the diets?
382-392 – increased methane is a wasteful process, meaning less energy is effectively used by the cow. Given you didn’t see a change in feed intake volume, milk yield (and you didn’t compare changes in liveweight) you are likely correct in stating that digestibility improved. However, given the global efforts to reduce methane emissions from agriculture would this be an acceptable trade off? I know you didn’t set out to conduct life cycle analysis, but do you predict that the emissions ‘savings’ from using broken rice as an alternative feed would be worthwhile? This warrants commenting on in this manuscript.
423-426 – in what species were each of these reporting?
436-437 – incomplete sentence? What is trying to be conveyed here?
General comments:
Low producing cows, is this reflective of the general herd in Thailand? Would you think these results would be similar in higher production animals?
It is oversight to not have analysed changes in liveweight. While the cows were not in early lactation it is possible that tissue accretion was occurring, this could have been observed within each period.
No comment on the technology used and its reliability? Given there is only one cited paper using this tech the reliability should be commented on.
English language is mostly excellent, some minor issues highlighted.
Author Response
Response to Reviewer 4 Comments
Comments and Suggestions for Authors
animals-2393384 "The energy contents of broken rice for lactating dairy cows"
This is a comprehensive and useful experiment. The information provided is generally sound and correct, although further details are required. The writing is generally excellent and the data is mostly discussed appropriately. Some changes can be made to improve the manuscript prior to publication. The information is highly relevant to the local and potentially international industries.
General comments:
Point 1: 66-68 – instead of repeating 30.9% perhaps reword to “replacing steam-flaked corn with steam-flaked brown rice (30.9%) did not alter milk yield but increased milk fat production” or similar
Response 1: Thank you for your suggestion. We made changes accordingly.
Point 2: 70-79 – while the information in monogastric species is interesting it is not very relevant to ruminants. Suggest condensing this section.
Response 2: Thank you for your suggestion. We made changes accordingly.
Point 3: 91-95 – what parity were the cows? Were they inseminated at any stage? Were cows milked in their pens or walked to a milking parlour? If walked, please describe distance etc.
Response 3: Thank you for your recommendation. Non-pregnant multiparous cows were used, and milk was in the metabolism pen. We added information accordingly.
Point 4: 96-100 – were all cows on the same diet within each period or was it one cow per diet per period? Please clarify.
Response 4: Yes, it was one cow per diet per period.
Point 5: Table 1 – be consistent with the number of significant figures reported in the footnotes (minerals).
Response 5: Thank you for your suggestion. We made changes accordingly, and the text now reads, “Minerals included 93.72 g/kg Ca, 46.86 g/kg P, 107.78 g/kg Na, 18.56 g/kg S, 8.24 g/kg Mn, 7.49 g/kg Zn, 3.37 g/kg Mg, 1.17 g/kg Cu, 0.15 g/kg Co, 0.01 g/kg K, 0.04 g/kg I, and 0.02 g/kg Se (Mineral #0106410029, Dairy Farming Promotion Organization of Thailand (D.P.O.), Saraburi, Thailand).”
Point 6: 128-134 – you weighed residual feed but it is not possible to accurately determine which portion of the diet was consumed given it is a TMR. This needs to be discussed as an experimental flaw.
Response 6: The total mixed ration was prepared using a horizontal TMR mixer, approximately 2,000 kg of dietary treatment ingredients mixtures were well mixed to ensure no sorted feeding, thus, no differences between feed offer and refusal composition.
Point 7: 135-144 – describe the met cages in more detail
Response 7: The text now reads, “The total collection technique was done in a digestion trial pen (165x375 cm) installed with a head cage respiratory gas system (105x80x173 cm) for each animal over five consecutive days [18]. The animals have relocated to a digestion trial pen on days 17 to 21 of each experimental period phase.”
Point 8: 146-165 – it is not possible from the description here or in the cited paper (Suzuki et al 2008) to adequately understand what the chambers / met cages were like. Were the animals able to lay down? Or were their heads in the ‘hood’ section for the whole period. What ambient temperature were the animals kept at? Were they indoors? What bedding (if any) was used? How was food delivered (in feeders, or on the floor?). was the feed in the hood area? Was the water also present in the hood area? If the animals were able to move their heads out of the hood, how many hrs a day were they actually sampled for?
Response 8: Thank you for your suggestion. We made changes accordingly, and the text now reads Animal Calorimetry. During the digestion trial, the oxygen consumption (O2), carbon dioxide (CO2), and methane (CH4) production of each animal were measured using an animal calorimeter, open-circuit indirect respiration calorimetry system, and the ventilated flow-through method using a head cage respiratory gas measurement [18]. The system consisted of a digestion trial pen, a head cage, a gas sampling and analysis unit, and a data acquisition and processing unit. A head cage is installed in front of the digestion trial pen and designed to be airtight, except for an air-inlet adjustable collar. Cows kept their heads in the hood section and had access to feed trays and automatic water for the day. Cows can lie down on rubber mate floor. A flow meter (NFHY-R-O-U, Nippon Flow Cell, Tokyo, Japan) was used to measure the flow rate and total air volume from the respiration chamber. The temperature and humidity of outflowing air were recorded electronically (ESPEC MIC CORP, Japan, model RS-12). The oxygen concentration in the inflow and outflow lines was measured using a dual-chamber paramagnetic oxygen analyzer (4100 Gas Purity Analyzer, Servomex Group, East Sussex, UK). Carbon dioxide and methane were also measured using an infrared gas analyzer (IR200 Infrared Gas Analyzer, Yokogawa Electric Co., Tokyo, Japan). The gas analyzers were calibrated daily against certified gases (Takachiho Chemical Industrial Co., Tokyo, JP); reference gases included two oxygen concentrations (19.0% and 20.6%), 1.89% carbon dioxide, and 1960 ppm methane. Calorimetric system recovery tests (98% to 104%) were conducted using the carbon dioxide injection method, by which a weighed amount of carbon dioxide gas was released into the system. The respiratory gas exchange measurements were taken at intervals of 7 min for 23.30 hours per day from the initial day at 08:00 am to 07:30 am the next day to determine energy partitioning and consumption over the last three days of the respiration collection period. Ambient temperature and relative humidity were not conditioned; their average was 27.7°C (23.3 to 36.6°C) and 82.7% (41.0 to 99.0), respectively. The metabolizable energy intake was calculated by deducting the urine and methane energy outputs from the digestible energy (DE) intake. Heat production was estimated using the Brouwer method (Equation 1), and the energy balance was calculated [19].
Point 9: 168 – do you mean “fan forced” oven? “until” reaching a consistent weight?
Response 9: Thank you for your suggestion. We made changes accordingly.
Point 10: 180 – do you mean fresh as in just mixed or fresh as in +7days after mixing?
Response 10: We measured the fermentation quality of the TMR silage fresh as in +7days after mixing (see Section 2.1).
Point 11: Table 3 – add hyphen in “nonfiber”
Response 11: Thank you for your suggestion. We made changes accordingly.
Point 12: Table 3, 4 & 5 – please clarify in the title that n=16 is n=4 per treatment group and not n=16 per group as the current formatting implies.
Response 12: We made changes accordingly, and the text now reads “n=4 per treatment”
Point 13: Tables – check formatting. It may be a PDF issue but it looks like some ‘tick marks’ or small borders are showing at the top and bottom of the tables.
Response 13: We made changes accordingly (Table 7 and 8).
Point 14: Table 5 – the “1” superscript is listed twice in the table. Please correct.
Response 14: We made changes accordingly.
Point 15: Table 6 – n = 15? Was one cow removed? Why? This needs to be described (this is also not mentioned later when n=15 data is presented). The results section should start with a statement as to why data from one animal was removed/not recorded etc. Also, update to be clear about n per group as described previously.
Response 15: Thank you for your suggestion. We added our result (Section 4.2) “Fifteen of a possible 16 energy balances were completed. During the collection of period 3, one cow became ill and was removed from total digestibility and respiratory gas collections.”
Point 16: 289 – “Methane energy lost cubically” due to what? Increasing BR content? Please clarify.
Response 16: Thank you for your suggestion. We added our discussion (Section 4.2) “Our result indicated that enteric methane emission energy loss (kJ/kgBW0.75) increased, suggesting the greater rumen fermentation of dry matter, especially on non-fiber carbohydrate and digestible fiber intake when including broken rice in the dairy cow diet.”
Point 17: Inconsistent use of BR abbreviation throughout, please be consistent.
Response 17: We made changes by spelling out all “BR” abbreviations to “broken rice” accordingly.
Point 18: Figure 1 – spacing/formatting of the data point symbols in the caption are not quite right (brackets partially covering the symbols). Please check.
Response 18: Thank you for your suggestion. We made changes accordingly.
Point 19: Table 7. Formatting – why is DE bold and underlined? Incorrect spelling “slope”, superscript 1 missing (not in the table but in the footnote),
Point 19: Response 1: We corrected accordingly.
Point 20: Table 8 – again why is DE bold and underlined?
Response 20: We corrected accordingly.
Point 21: 356-359 – this sentence is confusing and I cannot follow.
Response 21: We reworded, “Our data indicate that non-fiber carbohydrate intake substantially improved, suggesting a greater increase in total nutrients and energy supply to the lactating dairy cows when broken rice was included in the diet.”
Point 22: 378-380 – “Kendall et al. [31] indicated that the NDF intake was higher for cows fed 28% compared to 23% NDF in the diet, resulting in an increase in milk fat production.” – yes I would assume NDF intake was higher when cows were fed diets with higher NDF. I suggest rewordinig.
Response 22: We reworded, “Kendall et al. [31] indicated that higher NDF intake increased milk fat production in dairy cows. The results of this study indicated that milk fat yield depression might be caused by low fat or fiber intake, affected by rumen fermentation end products such as reduced acetate production as a precursor of milk fatty acid synthesis. Thus, depressing de novo fatty acid synthesis in the mammary gland [16,30].”
Point 23: 365-380 – Did you measure the PUFA levels in the diets?
Response 23: No, we did not analyze.
Point 24: 382-392 – increased methane is a wasteful process, meaning less energy is effectively used by the cow. Given you didn’t see a change in feed intake volume, milk yield (and you didn’t compare changes in liveweight) you are likely correct in stating that digestibility improved. However, given the global efforts to reduce methane emissions from agriculture would this be an acceptable trade off? I know you didn’t set out to conduct life cycle analysis, but do you predict that the emissions ‘savings’ from using broken rice as an alternative feed would be worthwhile? This warrants commenting on in this manuscript.
Response 24: Thank you for your suggestion; this is a good global impact issue. We reworded. “Our result indicated that absolute enteric methane emission energy loss (kJ/kgBW0.75) increased without alteration on methane yield (L/kg DM intake; data do not show) and methane intensity (L/kg Fat-and protein-corrected milk; data do not show), suggesting the greater rumen fermentation of digestible non-fiber carbohydrates and fiber when including broken rice in the dairy cow diet. In this study, daily methane emissions (258.7 to 303.6 L/d) were like previously reported [18,28] of lactating dairy cows, ranging from 329 to 360 L/d. Using broken rice in the diets of dairy cows would be an acceptable trade-off in reducing global enteric methane emission efforts.”
Point 25: 423-426 – in what species were each of these reporting?
Response 25: They are “broiler chickens”; we deleted L423-425 considering it irrelevant to ruminants.
Point 26: 436-437 – incomplete sentence? What is trying to be conveyed here?
Response 26: Thank you for your suggestion. We reworded it to “The net energy for lactation of broken rice in animal calorimetry has not been reported.”
General comments:
Point 27: Low producing cows, is this reflective of the general herd in Thailand? Would you think these results would be similar in higher production animals?
Response 27: It is generally low milk-producing cow herds in Thailand and other tropical developing countries. In Thailand, the average milk yield is approximately 12 kg/d. The limitations are heat stress and quality feed supplies. Feed intake is a limiting factor determining the nutrient and energy supply required for animal maintenance and productivity. Tropical feeding systems often rely on low-quality feed sources deficient in digestible nutrients and energy intake. Increasing the available nutrient supplies enables animals to improve derive milk production from carbohydrates, proteins, and fats.
Point 28: It is oversight to not have analysed changes in liveweight. While the cows were not in early lactation it is possible that tissue accretion was occurring, this could have been observed within each period.
Response 28: We measured live weight during the collection period in a short term in Latin square design; therefore, we did not report the body weight change.
Point 29: No comment on the technology used and its reliability? Given there is only one cited paper using this tech the reliability should be commented on.
Response 29: Thank you for your suggestion. We reworded it to (Section 4.3) “Our data suggested that in vitro had a greater variation in the estimation of net energy for lactation in dairy cows when compared with in vivo method. The limitation of animal calorimetry to determine net energy for lactation remained because of in vivo experiments require whole animal metabolism studies and respiratory gas measurements; they are time-consuming and costly and require many feed samples.”
Round 2
Reviewer 1 Report
Dear editor, I have already issued my opinion on this paper and I am not going to change it due to the serious flaws in methodology previously pointed out.
